# Discontinuous epidemic transition due to limited testing

Davide Scarselli [1,3], Nazmi Burak Budanur [1,3], Marc Timme [2] & Björn Hof [1✉]

High impact epidemics constitute one of the largest threats humanity is facing in the 21st century. In the absence of pharmaceutical interventions, physical distancing together with testing, contact tracing and quarantining are crucial in slowing down epidemic dynamics. Yet, here we show that if testing capacities are limited, containment may fail dramatically because such combined countermeasures drastically change the rules of the epidemic transition: Instead of continuous, the response to countermeasures becomes discontinuous. Rather than following the conventional exponential growth, the outbreak that is initially strongly suppressed eventually accelerates and scales faster than exponential during an explosive growth period. As a consequence, containment measures either suffice to stop the outbreak at low total case numbers or fail catastrophically if marginally too weak, thus implying large uncertainties in reliably estimating overall epidemic dynamics, both during initial phases and during second wave scenarios.

[1] Institute of Science and Technology Austria, Klosterneuburg, Austria. [2] Chair for Network Dynamics, Center for Advancing Electronics Dresden (cfaed), Institute for Theoretical Physics and Center of Excellence Physics of Life, Technical University of Dresden, Dresden, Germany. [3] These authors contributed equally: Davide Scarselli, Nazmi Burak Budanur. ✉email: bhof@ist.ac.at

For high impact epidemics such as the ongoing COVID-19 pandemic, countries at least initially rely on non-pharmaceutical interventions to slow the outbreak dynamics. Keeping the maximum number of simultaneously infected individuals sufficiently low is of paramount importance to not overload health care system capacities[1,2]. Testing, quarantining, and contact tracing have been combined with severe physical distancing measures across countries. Nevertheless, unlike, e.g., for the 2002–2004 SARS outbreak[3] and the 2013–2016 Western African Ebola virus epidemic[4] such combined countermeasures could not yet stop the present COVID-19 pandemic. It thus remains an open question how testing, quarantining, and contact tracing in combination with various physical distancing measures affect the epidemic dynamics, and in particular the epidemic peak that represents a worst case scenario regarding the pressure on the health care system.

Researchers and policy makers often implicitly assume that the peak, i.e., the largest fraction of simultaneously infected individuals, continuously varies with epidemic parameters and with the level of countermeasures implemented. In this article, we demonstrate that this fundamental assumption is incorrect once testing resources are limited. We reveal that the nature of the epidemic dynamics changes drastically from this naive picture and has unexpected, severe consequences. In particular, limited testing generically yields a discontinuous transition in the fraction of infected individuals in a population, a phenomenon dynamically accompanied by an interval of faster than exponential growth. Similar to related types of phase transitions in statistical physics such as discontinuous or explosive percolation transitions[5–8], limited testing effectively delays the transition, such that the fraction of infected individuals explosively becomes macroscopically large once effective epidemic parameters even only marginally cross a threshold. As a consequence, in the presence of limited testing, slight changes in countermeasures may induce huge macroscopic changes in the fraction of infected individuals, severely restricting the predictability of the epidemic transition. Discontinuous epidemic transitions have previously been pointed out in the context of limited vaccination supplies[9], limited control[10], and limited resources[11].

In many epidemic models, such as the Susceptible-Infectious-Recovered (SIR) model, the population is commonly considered large and divided into compartments such as susceptible ($S$), infectious ($I$) or recovered ($R$), and the evolution of these compartments is traditionally[12] modeled by ordinary differential equations (ODEs). For small populations, number fluctuations become relevant such that stochastic, microscopic network models are more appropriate, whereas for increasingly large populations, deterministic mean field approaches are usually considered suitable because relative fluctuations in the susceptible, infectious etc. populations become less and less important. In comparison to more complex models that take into account population structure and stochasticity, ODE models are often also motivated by the simplicity of implementation, as well as by the greater ease in analyzing and interpreting the results. With this perspective in mind, in large-scale outbreaks such as the ongoing COVID-19 pandemic, traditional ODE models would be expected to capture the overall general features of the epidemic dynamics.

As we explain below, if a disease spreads despite intervention, a large population size and a large number of tests do not constitute a sufficient condition for neglecting fluctuations. The key quantity during the early growth phase of an epidemic is the difference between two of these numbers, the number $N_T$ of daily tests and the number $N_S$ of individuals suspected to carry the disease and thus (ideally) to be tested. Regardless of the overall population size and other macroscopic population numbers, the difference $\Delta_{\text{Test}} = N_T - N_S$ may be or become small and thus introduces

relevant fluctuations. If it becomes negative the epidemic spread subsequently accelerates, i.e., during this phase the growth is faster than exponential. It is precisely this acceleration that fundamentally alters the nature of the epidemic transition and makes it discontinuous. As a consequence, a small variation of the epidemic parameters does not cause the expected small change to the growth process but yields disproportionate consequences with an explosive increase of the fraction of infected individuals in the population. The role of testing in inhibiting disease spread can be loosely considered analogous to pressurizing a fluid to hinder its vaporization. If the control parameter is too large, the situation will eventually get out of hand. For an epidemic, a reproduction number above one will eventually exhaust any finite test capacity. Once exceeded, the disease spread undergoes the aforementioned explosive phase and despite continued testing the reproduction number unavoidably increases towards its basic, uninhibited value. Likewise if heat is supplied to a pressure vessel at a too high rate, the increasing internal pressure eventually leads to the vessel's rupture and unavoidably a macroscopic fraction of the fluid vaporizes, essentially at once.

## Results

We will first illustrate the above mechanism in simulations of a basic dynamic agent-based model that simultaneously captures the epidemic dynamics and the influence of countermeasures and has previously been studied for analyzing the dynamics and control of Ebola[13]. Subsequently, we will show that the phenomenon is robust and hence independent of any details of the epidemic model used. For simplicity, we initially focus on two-dimensional square lattice grids where each agent represents an individual and interactions can be either short-range (via the four nearest-neighbor contacts) or long-range (see Fig. 1). The former represent fixed contacts, e.g., close family or colleagues, while the latter contacts are dynamically allocated (i.e., changed at each time step) and represent random encounters, e.g., during travel, shopping etc. An agent falls into one of four compartments,

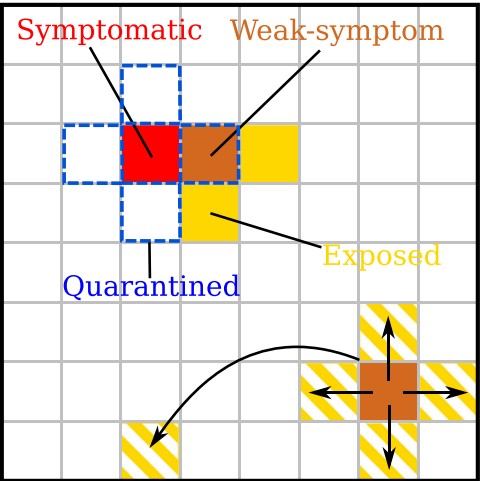

**Fig. 1 Spatial epidemic model with testing and quarantining.** The base model is illustrated on a square grid. Every day each infectious individual (agents are represented here as a tile on a lattice) interacts with their neighbors and with a randomly selected individual, and transmits the disease (arrows in figure) with constant probability if the individuals they interact with are susceptible. The tiles with yellow and white stripes denote the potential contacts that can be exposed. Upon identification of a positive case (red tile) all the neighbors are put into quarantine and tested (blue dashed border). Weak-symptom cases (brown tile with blue dashed border) can only be identified if they are neighbors of a known positive case.

susceptible ($S$), exposed ($E$), infectious ($I$), or recovered ($R$) resulting in an SEIR model. In addition, we split the infectious population into two categories, strong- and weak-symptom cases, where the latter represent individuals with either unspecific or no symptoms. As a result of intervention measures, an agent in each of the above states may or may not be quarantined, formally increasing the number of possible states to eight (for a detailed description of all the states and possible interactions see Supplementary Fig. 1). This basic model incorporates the main ingredients required to simulate epidemic outbreaks (i.e., it captures phase transitions to exponential disease spread), and has been shown to faithfully reproduce the course of the 2013–2016 Ebola outbreak[13].

Key features of the model can be understood from the example illustrated in Fig. 1. We consider discrete time dynamics with time steps representing days. Strong-symptom individuals (red) are immediately identified and automatically quarantined (blue dashed border). After testing positive, its four nearest neighbors are quarantined and queued for testing. For every new positive case, the quarantining and testing procedure is continued. In this simple scenario, all local contacts (four neighbors) are traced, however, prior interactions with distant sites are assumed untraceable. Weak-symptom cases (brown tile bottom right of Fig. 1) go undetected unless identified through contact tracing. At each time step, they can spread the disease with a constant probability to the four nearest neighbors plus to a randomly chosen distant site.

We start the simulations from a small number of weak-symptom infectious agents randomly scattered across a population of $P = 3162 \times 3162 \approx 10^7$. The incubation and infectious period are modeled with a Gamma distribution with parameters in the range of the ones reported[14] for COVID-19. The transmission probability is set to 0.28 to reproduce the average growth $\propto \exp(\kappa t)$ with the rate $\kappa = 0.3 \, \mathrm{day}^{-1}$ observed during the early exponential phase of the ongoing COVID-19 pandemic. In addition, we take 50% of the newly infected agents to show only weak symptoms. While the exact ratio of weak-symptom carriers of COVID-19 is unknown, their prevalence is reported in multiple studies[15–18]. Moreover, to allow for a scenario relevant to real testing we set an upper limit of daily tests of $10^{-4} P$ (i.e., 1000 tested individuals per day). This limit represents the largest fraction of the population tested in any European country during the first months of the COVID-19 outbreak[19]. In most countries, the daily tests conducted were significantly lower during the early phase of the epidemic but have since increased and at present are often larger. While focusing on the early stages the results reported below are robust against the specific values of $N_T$ as long as the daily test limit is significantly smaller than the total population size ($N_T \ll P$). Specifically, we show that all main features remain unchanged for a tenfold increase in test numbers, as well as for daily increasing and fluctuating test numbers, see Supplementary Fig. 6. The parameters chosen for the present study result in a basic reproduction number $R_0 \approx 3$. For large $P$ and $N_T$ an outbreak (leftmost curve in Fig. 2a) with such a high transmission rate can not even be halted by the highly efficient testing and contact tracing intervention scheme described above.

We next consider how the outcome of the epidemic is altered if the testing and contact tracing interventions are aided by additional mitigation measures (e.g., physical distancing). Unlike testing and quarantining that are simulated directly, additional mitigation measures are modeled by a reduction in the transmission rate. In order to investigate the response to different levels of mitigation the transmission rate is reduced, which translates to a continuous decrease in the basic reproduction number. As shown in Fig. 2a, the epidemic curve flattens at first continuously as the mitigation strength increases. However, once

the basic reproduction number marginally drops below a value of 2.5 the epidemic peak discontinuously drops to a very low value. Hence for $R_0 < 2.5$, the outbreak is halted (the fraction of infected tends to zero in the thermodynamic limit). The familiar continuous picture of flattening the curve is recovered when testing interventions are removed as shown in Fig. 2b. Here, a continuous reduction in $R_0$ causes the expected continuous decrease of the peak of the epidemic curve (see also Supplementary Video 1).

The discontinuity in the presence of testing and contact tracing is equally apparent when considering the total number $N_F$ of infected at the end of the epidemic, shown for decreasing mitigation strength (i.e., increasing $R_0$) in Fig. 2c. While testing and contact tracing can suppress outbreaks with basic reproduction numbers significantly larger than one, once containment fails it does so catastrophically, i.e., the fraction of the population eventually infected jumps from close to zero directly to a large fraction, in this case approximately $0.5 P$. The cause of the discontinuous response can be understood from the time evolution of the empirically computed[13] effective reproduction number $R_t$. As shown in Fig. 2d, for a suppressed outbreak ($R_0 = 2.3$, black circles) testing and contact tracing reduce the reproduction number to just below one and hence the number of infectious decreases exponentially and the outbreak is eventually suppressed. For $R_0 = 2.7$, however, the effective reproduction number can only be reduced to a value slightly larger than one. Consequently the number of infectious individuals increases exponentially.

So far the difference between these two cases is marginal, exactly as standard models would predict. As time proceeds however, in the latter case the number of suspects will eventually reach the test capacity limit, $\Delta_{\mathrm{Test}} = 0$ (at this point the positivity rate increases see Supplementary Fig. 2). Subsequently, a fraction of the infectious are only tested with a delay and therefore have a larger probability to transmit the disease. As shown in Fig. 2d, this leads to an increase in the reproduction number and hence the outbreak accelerates. Once set into motion, the number of unchecked suspects continues to increase and so does the reproduction number. Instead of the familiar exponential growth during epidemics, the growth at this stage is faster than exponential (see Supplementary Fig. 3) because $R_t$ and thereby the exponent of the growth dynamics, increases with time. A marginal difference in $R_0$ (compared to the suppressed case) is amplified into a significant difference in the effective reproduction number $R_t$ once the contact tracing capacity limit is exceeded. It is precisely this basic amplification mechanism that turns flattening the epidemic curve into a process with discontinuous overall outcome. This mechanism is independent of the epidemic model used. Exhausting test capacities during a high impact epidemic must necessarily lead to faster than exponential growth and an acceleration of the disease spread. While in practice this effect is not easy to identify because by definition test numbers are insufficient to capture the full spreading rate at this stage, faster than exponential growth phases have indeed been detected during the second wave of COVID-19 in various countries. For Italy, a European country exhibiting a clear and dominant second wave, a joint analysis of time series of new observed case numbers and of the positivity rate qualitatively agrees with our predictions, as illustrated in Fig. 3. In particular, a period of faster than exponential growth (Fig. 3a) coincides with an increase in the positive rate (Fig. 3b), supporting the hypothesis that limited tests are responsible for accelerated growth.

To further illustrate the robustness of the above results, we performed simulations in networks of varying complexity. First, a small world[20,21] network that includes agents of high connectivity while still keeping the simple grid structure. Next, beyond a basic

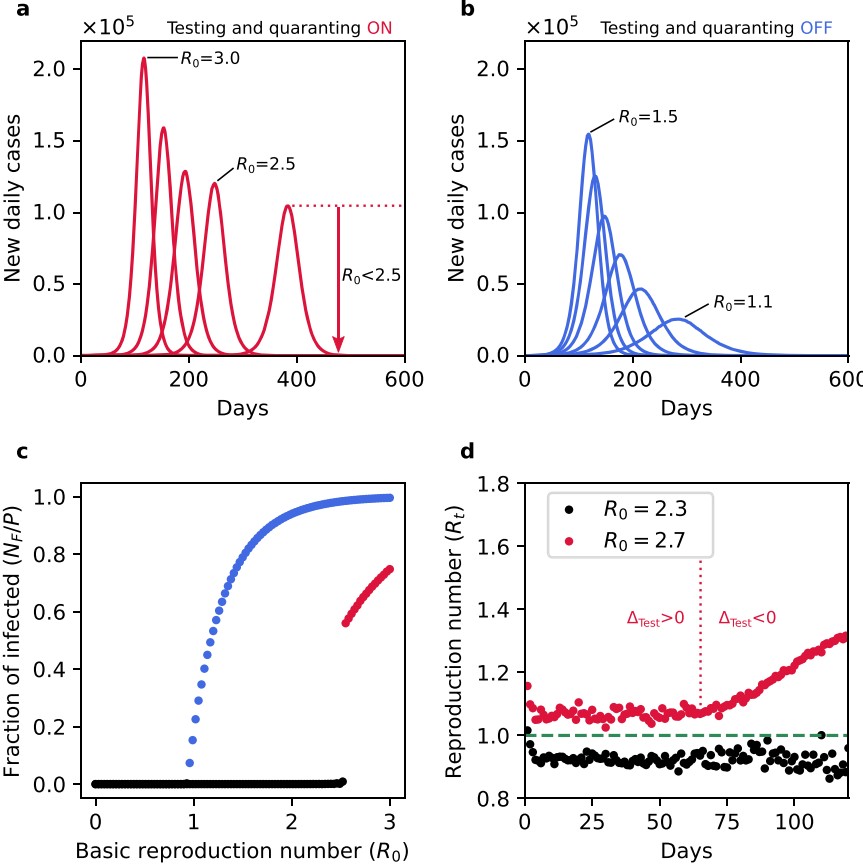

**Fig. 2 Discontinuity in flattening of epidemic curves. a** Daily new cases for continuously decreasing values of $R_0$ ($3 > R_0 > 0$), mimicking mitigation measures of increasing strength. Testing and quarantining are carried out at the same time with a capacity limit of $N_T = 1000$ tested individuals per day. Initially the peak reduces continuously in response to mitigation (red curves from left to right), however, once $R_0$ is reduced below 2.5 the epidemic curve drops to very small numbers of new cases (curves not visible in the figure scale). **b** Daily new cases for decreasing values of $R_0$ ($1.5 > R_0 > 0$) without testing and quarantining. Decreasing progressively $R_0$ gradually flattens the epidemic curve (blue curves from left to right) until a very low number of cases is reached (curves not visible in the figure scale). **c** Final fraction of infected ($N_F/P$) as a function of $R_0$ corresponding to the curves shown in **a** (red dots) and **b** (blue dots). The black dots denote outbreaks that have been effectively suppressed. When testing and quarantining are active the epidemic transition becomes discontinuous and happens at a higher basic reproduction number ($R_0 \approx 2.5$) in comparison to the usual continuous epidemic transition observed at $R_0 = 1$. **d** Evolution of the reproduction number with testing and quarantining active. Such measures efficiently reduce the reproduction number $R_t$ below unity for $R_0 < 2.3$ (black dots). For larger values of $R_0$ testing and quarantining can initially reduce the reproduction number to a constant level, however, $R_t$ remains above one (red dots). Owing to the continuing spread the number of suspects will eventually exceed the daily test limit and hence $\Delta_{Test}$ changes sign (red dashed line in **d**). At this point the spread accelerates and $R_0$ increases. The values of $R_t$ have been averaged over 400 and 100 simulations for $R_0 = 2.3$ and $R_0 = 2.7$, respectively. In all these cases the population size is $P = 3162 \times 3162 \approx 10^7$ people and epidemics start with 100 initial infectious.

grid, a scale-free network[22] and finally a scale-free network with additional random dynamic interactions (see Supplementary Fig. 4). We find that the discontinuity emerges across these various networks and is independent of the model complexity. As additional proof of the robustness of the results, we considered perturbations to the model parameters, domain sizes, and testing strategies (see Supplementary Figs. 5–7) and found consistent evidence for a discontinuous transition. Moreover, as no details about how the disease operates or spreads enter our qualitative modeling and analysis, all findings should be valid also beyond COVID-19 and in particular may support understanding and containing future pandemics. Together, these results strongly suggest that the induced discontinuous transition generally emerges once testing and contact tracing have an upper capacity limit.

A limited testing capacity does not only alter the response to mitigation during the early stages of an epidemic but equally introduces a discontinuity when considering lockdown scenarios of varying strength. For a simple illustration we again simulate an outbreak (at basic reproduction number $R_0 = 2$ in a population of size $P = 3162 \times 3162 \approx 10^7$) that has initially spread to $10^4$ infected

agents. The value of $R_0 = 2$ could be either interpreted as a less contagious disease, or as the same disease as in the previous case, where the reproduction number has been reduced by physical distancing measures. At this stage of the outbreak the number of suspects in the population already far exceeds the number of daily available tests ($\Delta_{Test} < 0$) and contact tracing cannot suppress the outbreak. To get the situation back under control, strong mitigation measures (i.e., a lockdown) are required and we consider that as a result the basic reproduction number is further decreased by $0 < \Delta R_0 < 2$, taking a lockdown period of 30 days for our illustrative examples (Fig. 4). For sufficiently strong lockdowns ($\Delta R_0 > \Delta R_c \approx 0.5$) the outbreak is eventually suppressed (i.e. the effective reproduction number is reduced below one). However, if the lockdown is just marginally weaker and $\Delta R_0 < \Delta R_c$ containment catastrophically fails: At the end of the lockdown the number of suspects still (marginally) exceeds the test capacity and as time proceeds $R_t$ increases (not shown). While in Fig. 2a the discontinuity separates epidemics subject to different mitigation levels, here the discontinuity arises from the difference in the number of active cases at the end of the lock down.

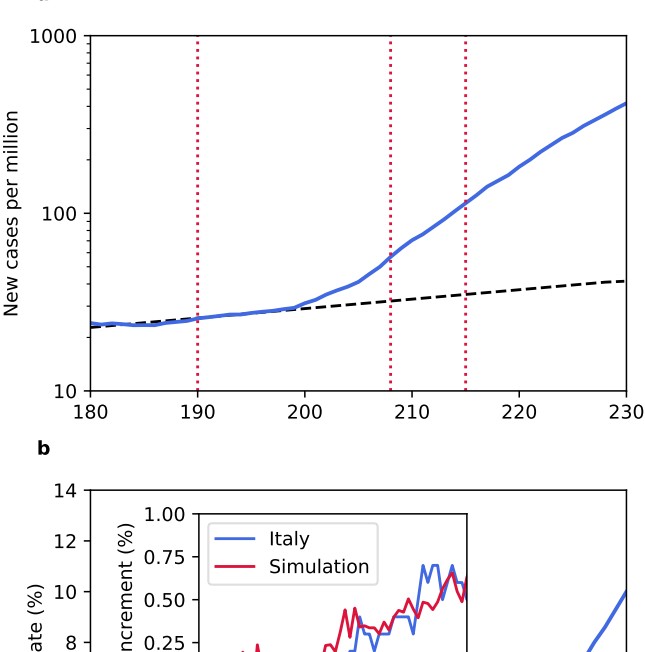

**a**

**b**

**Fig. 3 Period of faster than exponential growth during the second wave of COVID-19 in Italy. a** Number of new daily cases per million (logarithmic axis) reported in Italy. The red dotted lines indicate time intervals over which the new cases double, while the black dashed line represents an exponential fit to the data from day 180 to day 200. After a period of relatively weak exponential growth (days until 200) new cases surge suddenly leading to an intermediate phase of faster than exponential growth similar to what is observed in our simulations (cf. Supplementary Fig. 3). **b** The positivity rate, the fraction of tests that resulted positive in a given day, during the same time interval. As the number of new cases rapidly increases the contact tracing and testing strategy is put under strain and the positivity rate starts to increase. In the inset we compare the daily increment of positivity rate measured in Italy (blue curve) with the one predicted by a simulation (red curve). In both panels the data represent a 7-days moving average. The simulation parameters are the same ones used to generate the red curve of Fig. 2d.

In common epidemic models, testing and contact tracing are often incorporated in the basic reproduction number, yet that approach does not take into account capacity limits and hence cannot reflect scenarios in which such capacities are eventually exhausted as the epidemic continues to spread. As we have demonstrated, however, such capacity limits drastically alter the overall epidemic dynamics and thus need to be carefully considered both in research and for policy making. This holds in particular during early growth periods of the outbreak, as well as during potential second (or later) waves.

While during the COVID-19 pandemic the focus has been on the key role of mitigation in protecting the health care systems, the above results indicate that additional mitigation measures may play an equally vital role in protecting the efficiency of testing and contact tracing. If it fails, no matter how marginally,

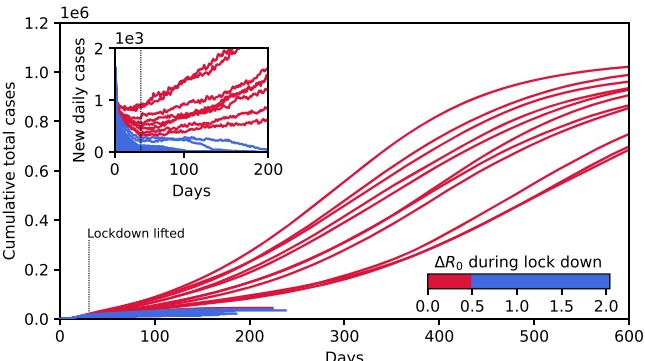

**Fig. 4 Discontinuous nature of lockdown scenarios.** Cumulative total number of cases for $R_0 = 2$ starting from $10^4$ infectious agents in a population $P = 3162 \times 3162 \approx 10^7$ and subjected to gradually stronger lockdowns. Mitigation measures are simulated by a reduction in the basic reproduction number in the range of $0 < \Delta R_0 < 2.0$. The duration is 30 days in all scenarios. Testing here is limited to $N_T = 1000$ individuals per day. Mild interventions ($0 < \Delta R_0 < 0.5$, red curves) only result in an initial drop (see inset) in daily new cases but ultimately cannot prevent a subsequent rise in numbers and eventually a high proportion of the population becomes infected. Stronger interventions ($0.5 < \Delta R_0 < 2$, blue curves) on the other hand efficiently bring the epidemic under control. Inset, corresponding epidemic curves of daily new cases. Reducing continuously $R_0$ during lockdown produces a family of epidemic curves that ultimately result in a discontinuous outcome: Either the outbreak is suppressed (blue curves), or containment fails catastrophically leading to a high proportion of the population being infected (red curves).

the disease begins to spread at an accelerating rate and in due course will likely hit a large fraction of the population. In practice, the eventual outcome of the epidemic might still be averted, if countermeasures are severely strengthened quickly after this acceleration. However, the suppression at this stage comes at a high cost and requires an extended lockdown to regain control. Our study suggests that a far better strategy is to use lockdowns as a preemptive tool. The faster than exponential growth phase should be avoided at all cost and hence additional measures must be implemented well before $\Delta_\text{Test}$ crosses zero, so that testing and contact tracing can work at their full efficiency throughout. Compared to the common practice of using a lockdown as the last resort, a preemptive strategy reduces the necessary economic downtime and saves human lives.

## Methods

**Spatial epidemic model.** Our base model is a spatial SEIR (Susceptible-Exposed-Infectious-Recovered) model, in which a population $P = 3162 \times 3162$ is represented by a two-dimensional grid where each grid point represents an individual. In addition to the above four compartments we distinguish between symptomatic ($I_S$) and weak-symptom ($I_W$) individuals, where the latter ranges from people who may have nonspecific symptoms (e.g., coughing) to entirely asymptomatic. With the introduction of intervention measures aimed at containing the disease spread, the individuals in the states $S$, $E$, $I_S$, and $I_W$ can be put under quarantine ($Q_S$ for $S$, $Q_E$ for $E$, and $Q_I$ for $I_S$ and $I_W$). All the eight states and the possible transition paths are shown in Supplementary Fig. 1 and described in the caption. Simulations start from a small group of 100 $I_W$ that are randomly scattered across the grid. Each infectious individual ($I_S$ or $I_W$) is assigned an infectious period, which is drawn from a Gamma distribution with mean 4 days and one day as scale parameter. During the infectious period these individuals can interact with each of the four neighbors and a randomly chosen additional individual. The disease is transmitted with a given probability, if the target individuals is susceptible (cf. Fig. 1). After the infectious period $I_S$ and $I_W$ transform into recovered ($R$) and can not interact any more with the population. Once a susceptible individual is infected, it transforms into exposed ($E$) and is assigned an incubation period, which is drawn from a Gamma distribution with mean 3 days and one day as scale parameter. After the incubation period is elapsed, the state of the individual is transformed from $E$ into $I_W$ with probability $p_W = 0.5$. For a large population size, $p_W$ thus represents the ratio of weak-symptom cases to the infected population. The transmission probability is

chosen to reproduce the average growth rate observed during the early exponential phase of the ongoing COVID-19 pandemic. To this end, we run several simulations without any containment and we pick the transmission probability that minimizes the difference from the growth $\propto \exp(\kappa t)$ with rate $\kappa = 0.3\,\text{day}^{-1}$.

**Testing and quarantining model.** The implementation of the epidemic mitigation is based on identification, quarantining and testing of suspect cases (cf. Fig. 1 for an illustrative cartoon of the process). The response starts when a first symptomatic case ($I_S$) appears and is recognized as a suspect case. The individual is immediately quarantined and tested. Upon the positive test result, the status is switched to $R$ and its neighbors are quarantined and queued for testing. Each day, $N_T$ (daily available number of tests) individuals in the queue are tested. The test outcome is revealed with a delay of one day and the same known positive cases can not be used more than once for tracing its neighbors. In case of negative test ($Q_S$) the individual is reverted to susceptible.

**Alternative networks.** The model can be easily extended to a different network structure while retaining the state transition rules and parameters. In order to assess the robustness of our results we considered three additional networks (the results are shown in Supplementary Fig. 4). For the first one, we chose Kleinberg's Navigable Small World as implemented in NetworkX 2.4 Python library. Here, each individual is connected to a random person on the grid, with the probability of being connected to a person decreasing as $d^{-2}$, where $d$ is the taxicab distance over the grid. Moreover, the connection is static and it is not assigned on a daily basis. In the second model we adopt a fully scale-free network with the number of connections per person drawn from a discrete zeta distribution with parameter 2 and cutoff 100, and all are static and do not change during the simulation. The third network is a scale-free network, but in addition every day each person is allowed to interact with an arbitrary random individual of the network. The resulting topology is dynamic and changes during the simulation.

## Data availability

The datasets generated and analyzed during the current study are available in the Zenodo repository, https://doi.org/10.5281/zenodo.4589567[23].

## Code availability

The codes used the network simulations are available in the Zenodo repository, https://doi.org/10.5281/zenodo.4589567[23].

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

## Acknowledgements

The authors thank Malte Schröder for valuable discussions and creating the scale-free network topologies. B.H. thanks Mukund Vasudevan for helpful discussion. The research by M.T. was supported by the Deutsche Forschungsgemeinschaft (DFG, German Research Foundation) under Germany´s Excellence Strategy–EXC-2068–390729961–Cluster of Excellence Physics of Life of TU Dresden.

## Author contributions

D.S., N.B.B., and B.H. designed the epidemic model with testing and quarantining. D.S. and N.B.B. wrote the Python code and performed the numerical simulations. D.S., N.B.B., M.T., and B.H. analyzed the data, interpreted the results, and wrote the paper.

## Competing interests

The authors declare no competing interests.
