## [Peer Review File · Nature Communications]

Reviewer #2 (Remarks to the Author):

The scenario of this work is the modeling of epidemic spreading dynamics and, in particular, of the COVID-19 pandemic. The authors show that when testing resources are limited the nature of epidemic dynamics changes and exhibits a discontinuous transition in the fraction of infected individuals and a subsequent super-exponential growth in the number of infections. The topic of the paper is certainly of interest especially in this year, where all world countries are facing with COVID-19 pandemic.

I enjoyed read this paper, since it is clear and well written. However, there are several critical points that I'd like to analyse.

1. To model the epidemics the authors adopted a SEIR model, a compartmental model that is an approximation of a real world epidemic spreading dynamics. Moreover, they simulate SEIR model on a square lattice grid, that is another rough approximation of the real world people contact network.

2. The simulation parameter are fixed by using empirical observations based on COVID-19 pandemic. It is not clear if and how results depend on these parameter, since there is no model parameters sensitivity analysis.

3. In the appendix (Fig. 4) simulations for small world and scale free network models are presented. However, in the case of small world network simulation parameter values (such as the population) are different respect the lattice and scale-free network simulations. Why?

4. It is unclear how the scale free network is generated. The authors wrote "In the second model we adopt a fully scale-free network with the number of connections per person drawn from a discrete zeta distribution with parameter 2 and cutoff 100, and all are static and do not change during the simulation.". But, what does it mean "fully scale-free"? What model do they use to generate such a networks? Barabasi-Albert model, configuration model, ...? And, again, how results depend on the value of parameters chosen? How many networks are generated and how the simulation results are obtained? An average of all the simulations? Moreover, any model used to generate a scale-free network is an approximation of a real world contact network and the level of accuracy strongly depends on the model used. So, I fear results can change by using different models or different combinations of parameter values.

5. All the simulation scenario (square lattice, small world and scale-free networks) are based on the strong hypothesis that all connections per person are static. This imposes a strong limitation to the study, since as is well known global mobility is one of the main drivers of the COVID-19 pandemic.

6. Simulation results are valid in the "thermodynamic limit" and this prevent the applicability of the results to the real world.

7. In all simulations the number of daily tests is considered constant, while this number is actually changing every day.

In conclusion, I think there is no scientific evidence that the results presented in this paper are applicable to real world pandemic such as COVID-19.

I fear that the conclusions of the authors are just a consequence of the models they adopted and are not valid in a real setting.

Reviewer #3 (Remarks to the Author):

This paper, inspired by current events, examines the behavior of a compartmental epidemic model (SEIR) under a testing and isolation policy. The main finding is that under limited testing capacity, we can potentially observe a discontinuous first order epidemic transition, in which the system may "jump" discontinuously from a healthy to a pandemic state. If I understand correctly, this offers a tradeoff in the potential merits of testing: On the one hand testing and isolation allows

suppression of the epidemic even if $R_0 > 1$. On the other hand, if the spread reaches a point in which testing cannot keep up, we risk a sudden unexpected burst in the spread, i.e. discontinuity.

The message is well written, timely and important. Hence, I am happy to review again a revised version of the paper. At the same time, the paper is currently a bit naive and narrow in its message - flaws that I think can be corrected with appropriate revisions. Hence, as the authors aim for a broad interdisciplinary journal, I strongly recommend the following improvements:

Narrative:

If my summary in the first paragraph is correct, I suggest to make it clearer in the paper's storyline. The tradeoff between the merits and risks of testing is not fully transparent in the paper's current presentation. It seems as if the authors claim that limited capacity testing is harmful for mitigation. This is, I think, not what they wish to convey. Rather, it is my understating that testing reduces the spread by effectively reducing R_0 . Only that what it fails, it does so dramatically, via a first order transition, hence it keeps the epidemic suppressed, but with the potential risk of a sudden outburst. If my reading is accurate, I suggest the authors sharpen this storyline and clarify it in their revisions. Indeed, as it currently reads this tradeoff is unclear. If, however, I misunderstood something... well, then, this - in and of itself - also suggests that the narrative has to be cleaned up.

Along these lines, it helps to have a simple illustrative analogy, that the readers can easily relate to. For example, you can think of testing as a way to suppress a pot of boiling water from turning gaseous. If sufficiently strong, such suppression can succeed. However, as the pot reached super-boiling temperature, if at some point, our suppression is not strong enough, you risk a sudden explosive transition. If this metaphor makes sense, I think it can provide a strong mental image to support the technical message.

The paper is almost solely focused on the specific mathematical result of the discontinuous epidemic transition, but only marginally discusses the broader implications, apart from the trivial - that testing capacity has to be high. For the statistical physics community this is indeed, an interesting result, however, the wider community of epidemiologists and public health experts, whom I think are integral to this paper's target audience, may not be as excited. I urge the authors to expand the breadth of their contribution and outline a set of relevant outcomes, beyond the mathematical results. I list some examples below, but I am open to alternatives suggested by the authors:

1. Estimate the required testing capacity for a given society. Or, alternatively, estimate the actual risks under a given capacity.
2. Are there any implications on improving testing policies, to gain the most under a given capacity.
3. If we are approaching the critical transition - is there a way to sense this. Specifically, since our "sensing" of the system is through testing, what will we observe in our testing close to criticality. Perhaps a sudden increase in the fraction of positive tests? Or some other indicator that our "pot" is about to burst?

Modeling:

The authors use a rather simple model for COVID-19, based on SEIR, allowing two separate compartments for mild vs. severe symptoms. The model is run on a Watts-Strogatz like network, i.e. lattice with occasional non-local interactions. This is sufficient for testing their theoretical predictions, but may not be enough to make specific claims on the current spread. Ref. 11 (Bar-On et al.) is a good reference from which to construct a more realistic disease cycle.

Along these lines, how sensitive are the results to changes in the network structure? I am pretty convinced that the first order transition will not be sensitive to that, but as stated above, to gain impact, the paper must push beyond the theoretical physics idealism, and provide some practical insights. These, may depend on details.

Estimating the testing capacity at 10^{-4} - is there a source for that? To me it seems that many countries have exceeded this figure. Am I missing something?

Results:

The paper mentions a super-exponential proliferation of the epidemic in several locations. Yet, such outbreak patterns are not shown in any of the main text figures. A single, and in my opinion, unconvincing example appears in the SI. To me it seems, that once $DT_{\text{test}} < 0$, we observe a sudden change in the pandemic spread, as the effective R_0 becomes larger. Hence the system transitioned from one exponential behavior to another - namely we see a shift in the slope in a semi-log plot. Super-exponential seems to imply something essentially different, e.g., $\exp(x^2)$, but I do not think that this is what we are actually observing.

I am interested to learn how the explosive transition observed in this model is related to other explosive transitions that are recently gaining interest in the complex systems community.

Figure 3. Would it be better to use a log scale for the y-axis? Also - I suggest to indicate the time when lock-down is lifted.

Technical:

Page 3, second paragraph: We model the incubation and infectious period with a Gamma distribution with parameters *are* similar to the ones reported for COVID-19. - parameter *that* are

Page 3, second paragraph: The mechanism and the induced discontinuous transition thus generally *emerges* if testing and contact tracing have an upper capacity limit. - *emerge*

The paper is rather poor in citations. Specifically, it ignores several recent contributions that are related directly to discontinuous transitions in epidemic spreading, testing policies and COVID-19 mitigation strategies. Providing more context (and credit) to earlier work is recommended.

General replies to the comments of reviewers.

We thank all reviewers for their thorough and detailed comments on the manuscript and their comprehensive evaluation of our work.

Reviewer 2 highlights precisely the main novel point of our work in demonstrating that “when testing resources are limited, the nature of epidemic dynamics changes and exhibits a discontinuous transition in the fraction of infected individuals and a subsequent super-exponential growth in the number of infections.” The reviewer finds that the topic is “[...] of interest especially in this year [...]” and that they “enjoyed read(ing) this paper, since it is clear and well written”. Similarly, Reviewer 3 evaluates our manuscript as “well written, timely and important.” and asks us to strengthen its presentation to underline its broader implications.

Both reviewers have questions, comments and suggestions to further improve the manuscript. As improvement, reviewer 2 wonders whether the discontinuity depends on details of the model or the thermodynamic limit of infinite populations, and as such may not be robust; reviewer 3 invites us to further sharpen the storyline to clarify the broader implications and believes that shortcomings in the original version can be overcome in our revisions to address the readership of a broad interdisciplinary journal. Moreover, the Reviewers (as well as the editor) ask us to think about where our model predictions might be visible in or relevant for real world epidemics.

Generality of findings: One main aim of our work, as reviewer 3 correctly identifies, is to point out that testing and quarantining may help to reduce case number growth but that if a limited testing capacity is reached, growth extremely rapidly accelerates. For reasonably large populations this finding implies an explosive transition from a moderate or small total number of cases to macroscopically large numbers (which may only be suppressed by rapid severe lockdown or vaccination if available). We would like to emphasize that these findings are completely general and reach far beyond the current COVID-19 pandemic. In addition, these findings require rethinking and modifying standard models of epidemic dynamics where testing and quarantining often is included only implicitly by reducing overall infectiousness parameters. Our results further indicate that models capturing the reported phenomenon need to explicitly set a threshold or adapt infectiousness parameters dynamically with case numbers.

Robustness: As we now elaborate more broadly, the discontinuity and the accompanying phase of faster than exponential growth is robust against varying a broad range of model parameters, both dynamical and structural (network topology) and is thus expected to be visible across vast families of models with explicit (possibly temporally varying) thresholds as well as across disease types in the real world.

Applications to COVID-19: As we also now illustrate with an example, time series of simultaneously observed case numbers and test positivity rate show qualitatively the same change in growth pattern as the one predicted by our work: from mild to rapid exponential growth. Easily observable quantities are, for instance, a decreased doubling time of case numbers in conjunction with a simultaneous(!) increase in positivity rate. Our results moreover suggest a key role for preemptive lockdown or other distancing measures.

We have now revised and amended both the manuscript and the Supplement with additional theoretical arguments, real data analysis and evaluated simulations to strengthen these points and more specifically highlight the breadth and applicability of our conceptual findings. We moreover address all remaining comments in detail in the point-by-point responses below.

Replies to the comments and questions of reviewer 2

We thank the referee for the careful reading of our manuscript and for the many helpful remarks and suggestions. We have addressed all points raised individually below and have revised the manuscript accordingly.

The scenario of this work is the modeling of epidemic spreading dynamics and, in particular, of the COVID-19 pandemic. The authors show that when testing resources are limited the nature of epidemic dynamics changes and exhibits a discontinuous transition in the fraction of infected individuals and a subsequent super-exponential growth in the number of infections. The topic of the paper is certainly of interest especially in this year, where all world countries are facing with COVID-19 pandemic.

I enjoyed reading this paper, since it is clear and well written. However, there are several critical points that I'd like to analyse.

1. To model the epidemics the authors adopted a SEIR model, a compartmental model that is an approximation of a real world epidemic spreading dynamics. Moreover, they simulate SEIR model on a square lattice grid, that is another rough approximation of the real world people contact network.

Response: The simulations are carried out on a variety of network models, grid based as well as scale free without a grid. Please note that it is not our intention to make a quantitative precise prediction for a real world situation but rather to report a robust, network independent phenomenon. Limitations in contact tracing lead to phases of faster than exponential growth and a discontinuity independent of network details.

2. The simulation parameters are fixed by using empirical observations based on COVID-19 pandemic. It is not clear if and how results depend on these parameters, since there is no model parameters sensitivity analysis.

Response: The parameter choice is of course motivated by the present epidemic, but the discontinuity reported does not depend on any of these parameters. The only necessary requirements for this discontinuity to occur is that there is a limit on the number of suspects that can be tested per day. If the reproduction number is greater than one, the number of suspects will eventually outgrow the number of available tests. The resulting accelerated, superexponential growth then necessarily leads to a discontinuity in the outcome when compared to situations where the reproduction number could be held marginally below one. The underlying mechanism is hence very basic and does not rely on any other conditions.

3. In the appendix (Fig. 4) simulations for small world and scale free network models are presented. However, in the case of small world network simulation parameter values (such as the population) are different respect the lattice and scale-free network simulations. Why?

Response: The choice of a smaller population for the Kleinberg Small World network simulation was dictated by memory requirements of the code that generated the network. We have now optimized the code and run the simulation on the regular domain size with the usual parameters and updated the SI.

4. It is unclear how the scale free network is generated. The authors wrote "In the second model we adopt a fully scale-free network with the number of connections per person drawn

from a discrete zeta distribution with parameter 2 and cutoff 100, and all are static and do not change during the simulation.". But, what does it mean "fully scale-free"? What model do they use to generate such a networks? Barabasi-Albert model, configuration model, ...? And, again, how results depend on the value of parameters chosen? How many networks are generated and how the simulation results are obtained? An average of all the simulations? Moreover, any model used to generate a scale-free network is an approximation of a real world contact network and the level of accuracy strongly depends on the model used. So, I fear results can change by using different models or different combinations of parameter values.

Response: The term fully-scale free is used to denote that in such network topology there is no local neighborhood, differently from the other networks considered. The network is generated by connecting each individual with other individuals randomly selected from the whole network. The number of these connections is drawn from a discrete zeta distribution with parameter 2 and cut-off 100 (note that for the Barabasi-Albert model the distribution has parameter 3, i.e. less connections per node on average). The infection rate is chosen always to give a basic reproduction number $R_0=3$.

In our revisions, we have studied the influence of changing various dynamical model parameters as well as network topologies. The results indicate strong robustness against such variations in both dynamics and topologies. Across all variations, the discontinuity of the transition robustly persists.

5. All the simulation scenario (square lattice, small world and scale-free networks) are based on the strong hypothesis that all connections per person are static. This imposes a strong limitation to the study, since as is well known global mobility is one of the main drivers of the COVID-19 pandemic.

Response: We thank the reviewer for raising this issue. Apparently there is a misunderstanding and we revised the manuscript accordingly. Although the lattice is static (or in any network, the number of local neighbors is fixed), all distant connections are chosen dynamically and change every day. One way of viewing this is that the static neighbors can be interpreted as family and, e.g., close friends while the random connections are encounters due to mobility. Either way the dynamic component does not affect the main result; as pointed out above a limitation in test capacity must necessarily result in phases of faster than exponential growth and a discontinuity. In addition we have also added a dynamic component to the scale free network simulation, see new SI Figure 4c, which in its original version (SI Figure 4b) had been static. Also here the dynamically chosen link does not alter the nature of the transition and the discontinuity is robust to any such changes. Of course the discontinuity will only become apparent if no further measures, e.g. lockdown, are taken. The faster than exponential growth on the other hand will robustly be found in practice (see below).

6. Simulation results are valid in the "thermodynamic limit" and this prevent the applicability of the results to the real world.

Response: We thank the reviewer for raising this point. Whereas the phase transition (as all phase transitions in all systems) indeed mathematically occur only in the limit of infinite system size, none of the model results presented are only valid in the thermodynamic limit. In fact, all model simulations by construction report results for finite population sizes and all show the discontinuity predicted (which, due to finite system size mathematically is "just" a

very large finite step as opposed to other finite steps that are orders of magnitude smaller.. In order to clarify better this point we added a figure to the SI where we compare two simulations in two population sizes that are different by a factor 10. The epidemic transition remains unaffected, provided that the initial number of infectious and test capacity is scaled accordingly. These results strongly indicate that the results very closely resemble those of the true phase transition that occurs in the thermodynamic limit. As a consequence, the proposed mechanisms applies equally to large finite systems as occurring in real populations

7. In all simulations the number of daily tests is considered constant, while this number is actually changing every day.

Response: Thank you for pointing this out and we now clarify that the discontinuity and faster than exponential growth phases are robust to temporal changes in test limits. For simplicity and to avoid confusion that the discontinuity may be caused by a temporally changing test number, in the original version we consciously chose to keep the number of daily test constant. To further clarify this point we created an additional figure (Supplementary Fig. 6) in which we consider a much higher testing capacity ($NT=10000$) and a linear, ramp-up testing scenario with strong daily oscillations (20%). The transition remains discontinuous, notwithstanding the larger amount of available tests, strongly fluctuations, and gradual ramp-up. The theoretical reason for the persistence is very simple, the number of cases grow exponentially and as long as the number of tests does not grow at the same exponential rate, eventually there will be more suspects than available tests. Crucially at this point contact tracing loses efficiency. In practice this loss of efficiency is measurable in an increase of the positive rate. This is precisely the stage at which the disease spread accelerates leading to the reported phenomenon.

In conclusion, I think there is no scientific evidence that the results presented in this paper are applicable to real world pandemic such as COVID-19. I fear that the conclusions of the authors are just a consequence of the models they adopted and are not valid in a real setting.

Response: We strongly disagree with this conclusion. We hope it has become clear from the added data that the reported phenomenon is robust and in particular independent of network structure and dynamical model parameters. By its very nature, a discontinuous phase transition is insensitive against parameter variations, including the addition of new parameters (which can be viewed as a variation of parameters away from zero). It does not rely on model choice, any type of symmetry or tuned parameters in any way. The only requirement for the accelerated growth to occur is that the number of suspects at some time during the growth phase exceeds the test number. Finally we would like to point out that the predicted accelerated growth is indeed observed in the current epidemic and it meanwhile has been pointed out that growth exceeded the exponential growth predicted by standard models.

As shown now in the main text (Fig. 3) for the example of Italy, the number of daily reported cases doubled in shorter and shorter intervals (Fig. 3a). At the same time when this super exponential growth occurred, the positive rate of testing increased, confirming that this event occurred when test numbers became exhausted (Fig. 3b), as our theory predicts.

Replies to the comments and questions of reviewer 3

We thank the referee for the careful reading of our manuscript and for the many helpful remarks and suggestions. We have addressed all points raised individually below and have revised the manuscript accordingly.

If my summary in the first paragraph is correct, I suggest to make it clearer in the paper's storyline. The tradeoff between the merits and risks of testing is not fully transparent in the paper's current presentation. It seems as if the authors claim that limited capacity testing is harmful for mitigation. This is, I think, not what they wish to convey. Rather, it is my understating that testing reduces the spread by effectively reducing R_0 . Only that what it fails, it does so dramatically, via a first order transition, hence it keeps the epidemic suppressed, but with the potential risk of a sudden outburst. If my reading is accurate, I suggest the authors sharpen this storyline and clarify it in their revisions. Indeed, as it currently reads this tradeoff is unclear. If, however, I misunderstood something... well, then, this - in and of itself - also suggests that the narrative has to be cleaned up.

Response: Indeed this is correct. We have now improved the narrative and already clarify in the abstract that testing strongly inhibits spread but when it fails it does so catastrophically.

Along these lines, it helps to have a simple illustrative analogy, that the readers can easily relate to. For example, you can think of testing as a way to suppress a pot of boiling water from turning gaseous. If sufficiently strong, such suppression can succeed. However, as the pot reached super-boiling temperature, if at some point, our suppression is not strong enough, you risk a sudden explosive transition. If this metaphor makes sense, I think it can provide a strong mental image to support the technical message.

Response: Thank you for this helpful suggestion. We have included the analogy to a boiling fluid, which although is not precise (in the fluid case the transition does not change from second to first order), it nevertheless is a very useful illustration.

The paper is almost solely focused on the specific mathematical result of the discontinuous epidemic transition, but only marginally discusses the broader implications, apart from the trivial - that testing capacity has to be high. For the statistical physics community this is indeed, an interesting result, however, the wider community of epidemiologists and public health experts, whom I think are integral to this paper's target audience, may not be as excited. I urge the authors to expand the breadth of their contribution and outline a set of relevant outcomes, beyond the mathematical results. I list some examples below, but I am open to alternatives suggested by the authors:

- 1. Estimate the required testing capacity for a given society. Or, alternatively, estimate the actual risks under a given capacity.*
- 2. Are there any implications on improving testing policies, to gain the most under a given capacity.*
- 3. If we are approaching the critical transition - is there a way to sense this. Specifically, since our "sensing" of the system is through testing, what will we observe in our testing close to criticality. Perhaps a sudden increase in the fraction of positive tests? Or some other indicator that our "pot" is about to burst?*

Response: Thanks for these very valuable suggestions. Given available time series, it seems easiest and also most sensible to directly study option 3 suggested by the reviewer: One observable indicator indeed is a sharp increase in the positivity rate. In the revised manuscript, we newly added the observed case number time series of Italy (as Fig. 3a) and the simultaneously observed positivity rate of testing (as Fig 3b of the main manuscript). The

comparison to our simulation results suggests qualitative (and rough quantitative) consistency. We remark that the qualitative agreement was expected from our theoretical and computational analysis, whereas the close quantitative agreement between our model prediction and the increase in the same quantity observed in Italy during October may be coincidence. This issue notwithstanding, the simultaneous occurrence of both positivity rate and rapid increase in case number growth is exactly as predicted by our arguments.

In addition, regarding relevant outcomes of our study, one very important lesson to be learned is that a lockdown should be used as a preemptive tool and should be implemented before the explosive / faster-than-exponential growth. Once the acceleration sets in, reducing numbers becomes prohibitively difficult and expensive. Authorities should hence react well before testing and contact tracing becomes overstrained, i.e. before Δ_{Test} crosses zero. At such an earlier stage a shorter lockdown will suffice.

Modeling:

The authors use a rather simple model for COVID-19, based on SEIR, allowing two separate compartments for mild vs. severe symptoms. The model is run on a Watts-Strogatz like network, i.e. lattice with occasional non-local interactions. This is sufficient for testing their theoretical predictions, but may not be enough to make specific claims on the current spread. Ref. 11 (Bar-On et al.) is a good reference from which to construct a more realistic disease cycle.

Response: The model can be easily made more complex (e.g. adding pre-symptomatic transition, hospitalization, limited immunization etc.). However at the present stage we would prefer to keep the focus of the manuscript on the basic mechanism and this can be much more readily understood for the base model. We have however added real world observations of the predicted explosive spread and we illustrate that indicator for this to occur (change in positive rate) accurately matches between our model and in the real world example (Italy in this case). We hope that this addresses the reviewers concern, otherwise we can in addition include a more complex model structure.

Along these lines, how sensitive are the results to changes in the network structure? I am pretty convinced that the first order transition will not be sensitive to that, but as stated above, to gain impact, the paper must push beyond the theoretical physics idealism, and provide some practical insights. These, may depend on details.

Response: Thank you for this question. We have now revised the manuscript and Supplement to strengthen this point. We have increased the population size in the small world and scale free network. We altered the scale free network and included daily random interactions to make it somewhat more realistic (see Supplementary Fig. 4). We also tested the robustness of the results to variations of model parameters, namely the number of initial infections, the infectious and exposed period, and the ratio of weak-symptom cases. The results are shown in Supplementary Fig. 7 and further confirm the robustness of our findings. The acceleration of the spread and the discontinuous nature of the transition persist throughout.

In summary, the qualitative results, a discontinuous jump and two vastly different outcomes below and above the threshold set by testing (as expected) are found to be very robust against changes to various aspects of the model dynamics and of the underlying network topology. Also as expected, the results vary quantitatively with varying parameters, dynamics or topology, yet surprisingly little.

Estimating the testing capacity at 10^{-4} - is there a source for that? To me it seems that many countries have exceeded this figure. Am I missing something?

Response: The simulations were carried out during late spring when numbers in Europe were at this level (e.g. see test numbers in Austria during March). Indeed since then test numbers have considerably increased. This however does not change the nature of the transition. We have carried out additional simulations that mimic more realistic scenarios. In one case we have a much higher constant number of tests (10000, or $10^{-3}P$), while in the other one we considered a linear increase from 100 to 10000 with random fluctuations. In both cases a discontinuity is found, see Supplementary Fig. 6. Please also note that testing in our simple model is far more efficient, as all close contacts are known and no tests are wasted. Moreover, any individual requires only a single test and results are known in one simulation day.

Results:

The paper mentions a super-exponential proliferation of the epidemic in several locations. Yet, such outbreak patterns are not shown in any of the main text figures. A single, and in my opinion, unconvincing example appears in the SI. To me it seems, that once $DT_{test} < 0$, we observe a sudden change in the pandemic spread, as the effective R_0 becomes larger. Hence the system transitioned from one exponential behavior to another - namely we see a shift in the slope in a semi-log plot. Super-exponential seems to imply something essentially different, e.g., $\exp(x^2)$, but I do not think that this is what we are actually observing.

Response: We have added an illustration of the phase of faster than exponential growth as Fig. 4 to the main paper and the comparison to real world data illustrates that, as pointed out, this is an intermediate adjustment phase. However, this is not a sudden switching from one exponent to another but a gradual, continuous increase of the exponent. In case the referee may nevertheless find the “super exponential” term misleading we have exchanged it to ‘faster than exponential’.

I am interested to learn how the explosive transition observed in this model is related to other explosive transitions that are recently gaining interest in the complex systems community.

Response: Changing a process from exhibiting a continuous phase transition to exhibiting an explosive one (continuous or discontinuous) generally involves the delay or shift of the transition point (in parameter space).

For the paradigmatic example of explosive percolation (the transition to macroscopic connectedness upon adding links to existing nodes of a network), the critical number of links per node above which the percolation order parameter (the size of the largest cluster, the largest connected component relative to system size) starts growing from zero is larger than the critical (relative) number of links in standard (e.g. random network) percolation. In the case of explosive percolation, the transition point is shifted because the percolation rules of local “interactions” (the conditions under which links add) hinder the growth of the largest cluster as new links preferentially add to smaller clusters, creating multiple large clusters until the largest cluster becomes macroscopic (essentially by adding one or a few other large clusters, thereby creating an “explosion” rather than a slow growth).

For epidemic dynamics, our results in the manuscript and additional preliminary theoretical studies (in progress) suggest the following similarity of delaying the transition. If a model explicitly contains a testing threshold (not only effectively by lowering epidemic standard infectiousness constants), as we propose in the current manuscript, the tests help tracing and quarantining individuals that otherwise would spread the disease. They thus result in low(er) cumulative case numbers even at $R_0 > 1$ at which case numbers would rapidly exponentially grow if no testing and quarantining were present. However, if the number of

tests are limited and at high case numbers becomes insufficient to identify most individuals capable of spreading the disease, the effective infectiousness suddenly becomes large. Once this point is crossed (in time or in R_0), the case numbers start growing (approximately) at a rate defined by the process without testing (and thus lower infectiousness). As a consequence, the cumulative total case numbers explosively grow from being small (essentially zero relative to the population size) to macroscopic, i.e. proportional to population size.

In the manuscript, we briefly mention that the explosive transition results from a delay of the transition point. A more detailed and more theoretical study is ongoing.

Figure 3. Would it be better to use a log scale for the y-axis? Also - I suggest to indicate the time when lock-down is lifted.

Response: We decided to use a linear scale to highlight the dramatic difference between successful (blue) and ineffective (red) lockdowns. Compared to log-scale, the difference is visually clearer in linear scale plots because linear scale delineates microscopic total case numbers from macroscopic ones (proportional to total population size in the thermodynamic limit), see figure below. Moreover, our model (and in fact, no generic model) is capable to predict exact case numbers, so these are not helpful to a reader. We welcomed the suggestion to indicate the time when the lockdown is lifted and accordingly corrected the respective Fig. 2d in the main text.

Technical:

*Page 3, second paragraph: We model the incubation and infectious period with a Gamma distribution with parameters *are* similar to the ones reported for COVID-19. - parameter *that* are*

*Page 3, second paragraph: The mechanism and the induced discontinuous transition thus generally *emerges* if testing and contact tracing have an upper capacity limit. - *emerge**

The paper is rather poor in citations. Specifically, it ignores several recent contributions that are related directly to discontinuous transitions in epidemic spreading, testing policies and COVID-19 mitigation strategies. Providing more context (and credit) to earlier work is recommended.

Response: We thank the referee for pointing out these required improvements. We have added further references about other causes for discontinuities in epidemic modeling, delineated those causes from the one we propose and analyze, and corrected the typographical errors.

Reviewer #2 (Remarks to the Author):

The paper has been substantially improved and the authors addressed all my concerns. I believe the paper is now suitable for publication in the journal.

Reviewer #3 (Remarks to the Author):

Well done.

I think the paper is now suitable for publication.

I wish the best of luck to the Authors.